# A Business Model Framework to Characterize Digital Multisided Platforms

**Marco Ardolino** **, Nicola Saccani, Federico Adrodegari * and Marco Perona**

Department of Mechanical and Industrial Engineering, University of Brescia, Piazza del Mercato, 15,
25121 Brescia BS, Italy; marco.ardolino@unibs.it (M.A.); nicola.saccani@unibs.it (N.S.);
marco.perona@unibs.it (M.P.)
* Correspondence: federico.adrodegari@unibs.it

**Abstract:** Businesses grounded upon multisided platforms (MSPs) are found in a growing number of industries, thanks to the recent developments in Internet and digital technologies. Digital MSPs enable multiple interactions among users of different sides through information and communication technologies. The understanding of the characteristics and constituents of MSPs is fragmented along different literature streams. Moreover, very few empirical studies have been carried out to date. In order to fill this gap, this paper presents a three-level framework that describes a digital MSP. The proposed framework is based on literature analysis and multiple case study. On the one hand, the framework can be used to describe MSP as it provides an operationalization of the concept through the identification of specific dimensions, variables and items; on the other hand, it can be used as an assessment tool by practitioners, as exemplified by the three empirical applications presented in this paper.

**Keywords:** multisided platforms; business model; descriptive framework; multiple case study

## 1. Introduction

Digitalization is transforming the competitive landscape, challenging incumbent firms and leading to business model (BM) innovations [1]. In recent years, there has been a sharp diffusion of web-based businesses that enable and facilitate demand matching, the so-called digital multisided platforms (MSPs) [2]. These businesses are characterized by the presence of a focal firm, the platform manager, providing the infrastructure that enables interactions and transactions among the users of two or more sides [3–7]. Examples of MSPs are online marketplaces of products and services [8], mobile software applications [9], social networks [10], crowdsourcing [11], dating [12] and job-seeking platforms [13].

MSPs have disrupted competition in industries. For instance, the nature of exchange in retail businesses has changed as the introduction of digital marketplaces has enabled the direct connection between customers and independent suppliers [14], shifting the inventory risk from the retailer to the supplier. Moreover, the development of a digital MSP (DMSP) is always characterized by the creation of peculiar networks and ecosystems which facilitate social interactions and in particular value cocreation and coinnovation [15–17]. Even though the most well-known platforms have been introduced in "pure service" B2C contexts, manufacturing companies are also developing digital MSPs aimed at collecting and processing data from installed base to provide advanced services like predictive maintenance or "pay per use" models [18,19].

Although multisided platforms have been discussed in the literature, little research has dealt with the characterization of this type of business [20,21]. Most studies develop analytical models focusing on a specific feature of MSPs, such as price structure, network externalities or competition (e.g., [4,14,22,23]), while a holistic approach to characterize the MSP is lacking. In addition, in spite of

the great diffusion of companies grounded on DMSP model, very few studies investigate these kinds of businesses empirically either through a single case study (e.g., [24,25]) or multiple case studies [26].

In order to fill those gaps, this paper develops a three-level framework for the characterization of MSPs. The development of the framework is based on the literature (literature analysis) and on a multiple case study (preliminary study), aimed at identifying the possible configuration options for the framework variables. The application of the complete framework to three case studies is also illustrated in the paper (main study).

The purpose of the framework presented in this paper is twofold. On the one side, it has a descriptive purpose and it may be used to identify the variables which characterize MSPs. This can certainly help traditional businesses to evaluate which are the aspects to be evaluated for a business model innovation towards a DMSP. Indeed, the developed tool thus allows practitioners to make in-depth reflections on their business model by evaluating any changes in the structure of one or more characterizing dimensions.

On the other side, practitioners can use the framework as an assessment tool to benchmark with competitors.

The paper is structured as follows: Section 2 provides the conceptual background of the research. Section 3 illustrates the objectives and the methodology. Section 4 presents the three-level platform framework, while Section 5 presents the empirical application of the framework. Section 6 discusses the empirical findings, while conclusive remarks and directions for future research are drawn in Section 7.

## 2. Background

### 2.1. Perspectives on the Platform Concept

Research on platforms has progressed simultaneously in different directions. Piezunka [20] points out three main research domains where the concept of platform has been adopted, namely: new product development, technology management and industrial organization. A brief description of the three perspectives is provided in Table 1.

The first research stream builds on several classic product development studies. According to this stream, a platform is a set of subsystems and interfaces characterized by a common structure from which a company can efficiently develop and manufacture a family of products [27]. A notable example of the application of this concept is Sony's Walkman in the 1980s, whose modularity has provided a way to increase scalability for the company [28].

The second perspective draws on research related to technology management. The key concept of this research stream is the industry platform, that is a product, service or technology which serves as a foundation upon which other firms can build complementary products, services or technologies [5,29]. The Intel microprocessor represents a typical example of industry platform. Indeed, the company focused on the control of microprocessors' architecture, while giving away to other actors in the complementary markets the opportunity to develop compatible connectors (e.g., chipsets, motherboards) and applications [30]. This perspective has its roots in the concept of open innovation, as it is based on the idea that creation of new products or services might effectively come from external resources [15]. Platform businesses also generate ecosystems capable of amplifying innovative power, thanks to external knowledge, and providing improvements both locally and globally [31].

Finally, the third perspective is related to the industrial organization stream. In this context the platform is intended as a business intermediating two or more distinct groups of users, generally matching supply and demand and enabling interactions among them. This stream uses the term multisided platforms to differentiate them from one-sided platforms, where the business enables interactions between users of one group only [32].

**Table 1.** The platform concept in different research domains.

| Research Domain | Concept | Perspective | Example | References |
|---|---|---|---|---|
| New product development | Product platform | The platform is a product that meets the needs of a core group of customers but is designed for easy modification into derivatives through the addition, substitution or removal of features. The platform allows to save costs and increase efficiency in product development through the reuse of common parts and ease in the manufacturing of a large number of derivative products. | Sony Walkman | [20,27,28] |
| Technology management | Industry platform | The platform is a product, service or technology that is developed by one or several firms and that serve as the foundation upon which other firms can build complementary products, services or technologies. | Intel microprocessor | [5,29,30] |
| Industrial organization | Multisided platform | The platform intermediates between two or more distinct groups of users enabling interactions among them. Therefore, a multisided platform consists of a shared facility in which the interactions take place among the users. | E-bay | [3,4,20,32,33] |

*2.2. Digital Multisided Platforms*

The concept of multisided platform has been extensively debated particularly in the information systems and management literature but only in recent years has the subject been approached with the logic of BM [26]. Scholars often use the term MSP interchangeably with other ones such as multi-sided markets [4,33], platform-based markets [34], platform ecosystems [19,35] and marketplaces [26].

The disruptive success of digital platforms such as Airbnb and Uber have given great popularity to this BM [36]. Multisided platforms, however, have existed for a long time: traditional "brick and mortar" shopping malls, for instance, bring together retailers and shopper, providing the physical facilities and services such as maintenance operations for retailers and babysitting for shoppers' children [37]. However, internet, mobile and digital technologies have been the catalyst for the massive diffusion of digital multisided platforms [16], increasing the reach of connecting platform sides, improving match-making mechanisms, enabling more efficient transaction management and more effective trust-building [2]. Furthermore, the combination of the platform model with Internet technology greatly improves the perceived utility and ease of use, ensuring better satisfaction as well as greater security and transparency [38]. Amazon, for example, from a pure retailer business, has moved to a MSP model over time, opening its business and combining its own inventory with that of independent suppliers [8]. Based on the literature, in this paper we define a multisided platform as a BM [7,39] that is:

1. based on the presence of a virtual or physical place (the "platform") which enables and facilitates the interactions between two or more different groups of users [3,33,40,41];

2.　characterized by interdependent relationships among the sides, because of the presence of indirect and bilateral positive network effects [8];

3.　potentially able to track the interaction events between the users involved [42].

This paper addresses digital multisided platforms i.e., MSPs based on digital and internet technologies.

## 2.3. Characterizing a Digital Multisided Platform

The literature helps to identify the main characteristics of a digital multisided platform (DMSP). First of all, multisided platforms differ from traditional BMs because of the presence of interactions and transactions among users of two or more participating sides [43,44]. Therefore, it is fundamental to implement a structure able to maximize the size of the sides [45]. Typically, it is possible to identify a supply side, the group of users offering or providing a product or a service, and a demand side, consisting of the users who use or benefit from the contents (product or services) provided by the supply side [10,46,47]. A critical role in MSPs is the platform manager, who mediates users' interactions and serves as the users' primary contact point with the platform [46]. In addition, advertisers may be present as a third side involved [48,49]. DMSPs also have peculiar pricing and revenue models. Since there are two or more sides involved, the platform manager needs to deal with the pricing issue of whether and how much the different sides charge [40].

Another peculiar aspect that differentiates MSP from traditional businesses is competition. Platforms are in fact affected by two distinct types of competition [50]. Inside competition occurs between users inside the same platform side, such as the competition among the different sellers for gaining customers in a marketplace. In turn, "outside competition" instead occurs among platforms as they compete to get the users on board their platform.

The literature investigated several aspects of MSPs, including network effects, pricing, integration and control, engagement, competition, advertisement and regulation and antitrust. Table 2 summarizes the main issues addressed by the literature in each area.

**Table 2.** Investigation areas about multisided platforms.

| Area | Description | Reference |
| --- | --- | --- |
| Network effects/Network externalities | Network effects (or network externalities) are a distinctive feature of a multisided platform, arising when the growth in usage by one side increases the value for the other side. Such interdependencies often lead to a feedback loop in which the number of participants on both sides affect each other recursively. | [3,4,40,50,51] |
| Pricing | The pricing structure in a MSP generally makes the revenue model very complex. For instance, the price for one side can be zero and the profit by the platform manager is made only on the other side(s). Moreover, platform managers can choose among different kinds of fees to be applied. | [3,10,44,51] |
| Integration and control | Platforms involve several users on each side. It is important to control the behavior of the users participating in the platform and contents provided. | [14,22,33,35] |
| Engagement | Due to the presence of network effects (externalities), attracting users presents peculiar challenges, differently from traditional businesses, such as the so-called chicken and egg dilemma. Users on side "A" would not participate without users on side "B" and vice-versa. It is important to implement appropriate strategies to incentivize participation to all the sides of the platform. | [48,50,52] |

**Table 2.** *Cont.*

| Area | Description | Reference |
|---|---|---|
| Competition | Platforms are subject to two main forms of competition: (1) outside competition—competition of the platform business with other similar businesses; (2) inside competition—competition among the users participating in the same side. | [53–55] |
| Advertisement | Advertisement is an element often present in platform-based businesses. On the one hand, it is an important source of revenue for many businesses. On the other hand, it could be counterproductive because it causes frictions in interactions among the sides. | [48,49,56] |
| Regulation and antitrust | Multisided platforms can provide new alternatives to conventional BMs and that tests the limits of existing regulatory policies. Traditional businesses and policy makers generally wonder and debate if MSPs conform to regulations and antitrust laws. | [56–58] |

The literature contributions mainly concern the application of analytical models aimed at investigating specific aspects in the areas identified in Table 2, e.g., equilibrium, allocative efficiency and competition between two different multisided platforms [59,60]. Studies holistically addressing the MSP configuration are lacking. Quite surprisingly, the literature also lacks detailed empirical studies that analyze the configuration and features of existing DMSPs, in order to compare them and define archetypes [21].

Osterwalder and Pigneur [39] have used the BM Canvas to describe a MSP. The Canvas has also been used by Muzellec, Ronteau and Lambkin [52] to develop a model of the evolution of the marketing strategies of two-sided Internet businesses. Moreover, it has been adopted by Wang, Tang and Jin et al. [10] to investigate the dynamics of revenue streams in mobile social networks. However, the BM Canvas has some limitations related to its general purpose as it lacks the focus on peculiar aspects of MSP, such as network externalities and the relationships among the sides involved, as well as between each side and the platform manager. More specifically, the BM Canvas does not encompass the features to describe the interactions between the users, that are instead merely considered as customer segments.

An original framework for the analysis of MSPs has been proposed by Raivio and Luukkainen [61], even though it does not analyze the characteristics of the sides and the configuration options. Furthermore, that model has not found applications neither in the literature nor in practice. Finally, Täuscher and Laudien [26] develop an original integrative framework, used for analyzing 100 MSPs in order to develop an empirically grounded taxonomy but it focuses only on marketplaces.

## 3. Research Objectives and Method

Given the paucity of studies investigating the main constituents of a DMSP, the research focus of this paper is on exploration and theory building; the paper develops and presents a framework encompassing the dimensions and variables to be analyzed in order to characterize and categorize DMSPs.

Qualitative research is therefore particularly appropriate [62] and the research process adopted can be ascribed to "iterative-grounded" theory [63], as done by other works in the field of MSPs such as the one of de Oliveira and Cortimiglia [16]. This method combines empirical data with the relevant literature in order to fill in the gaps. This procedure has allowed for evaluation of the features characterizing a MSP and has revealed aspects neglected by the literature. The proposed framework is based in fact on the combination of a literature analysis and data about representative cases of

DMSPs, expanding the corpus of knowledge on the elements characterizing a multisided platform and their options.

Figure 1 illustrates the three research steps and how they contributed to the outcome of this study.

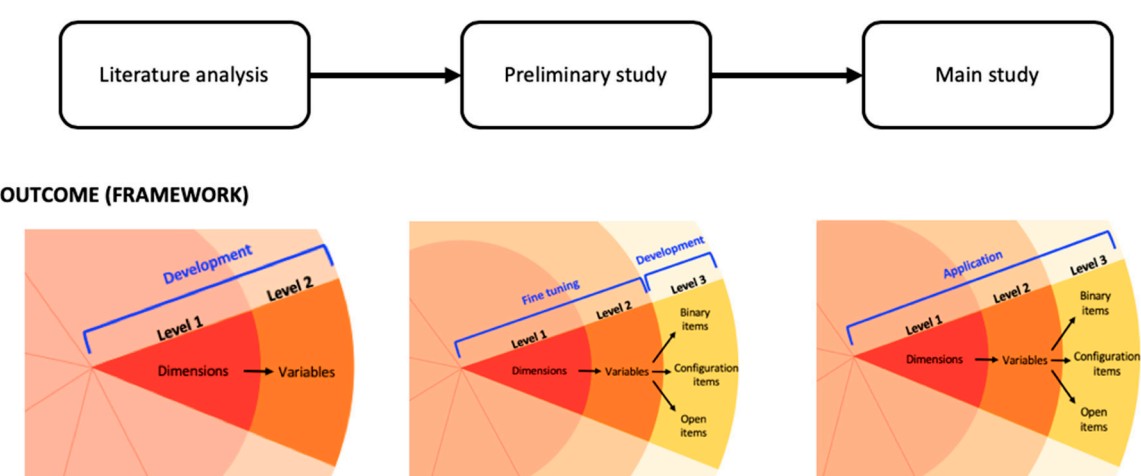

**Figure 1.** Platform descriptive framework.

## 3.1. Initial Framework Development

To point out the key research topics and findings about digital MSPs we searched the databases Scopus and WoS, through the keywords "multisided platform", "digital multisided platform" and their variations. The contents emerged from the literature have been codified and grouped into thematic categories (see Table 2). The results of this analysis provided the inputs for the identification of the dimensions and variables of the first two hierarchical levels of the framework (Figure 1).

For developing the framework, we adopted the morphological analysis method. Morphological analysis can help to identify attributes and specifications of a specific object of interest [26,64,65]. Witt, Stahlecker and Geldermann [66] acknowledge the creative nature of this technique since it can also be used to figure out new configurations that have not yet been adopted. Thus, we developed a framework characterized by a structure made of morphological boxes [67]. Each morphological box represents a specific constituent of the MSP BM and it is characterized by a number of variables and items. In this way it is possible to explore all the alternative configurations for a MSP BM.

## 3.2. Exploratory Case Study and Framework Refinement

In order to complement and refine the results of the literature analysis and the initial framework development, we conducted an empirical analysis which was characterized by two distinct processes: an exploratory and a main study [68]. The exploratory study allowed for further development of the framework, through the identification of the third-level items (Figure 1).

The exploratory study sample consists of 26 DMSPs. The case selection was based on the following criteria: (a) adequate coverage of the functions enabled by a multisided platform; (b) adequate maturity level (companies which are at least three years old have been selected) and (c) availability to share information and participate in the study.

The main information about the cases is summarized in Table 3. Company names are not reported for confidentiality reasons.

**Table 3.** Case studies description.

| Case | Description | Role of Person Interviewed |
|---|---|---|
| Case 1 | Crowdsourcing graphic design company | Vice President Engineering |
| Case 2 | Online marketplace for renting vacation homes | Host Operations Lead |
| Case 3 | Classifieds operating in second-hand cars and vehicles industry | Founder |
| Case 4 | Booking platform for hotels and other kinds of accommodations | Regional Director |
| Case 5 | Food delivery platform | Analytics Manager |
| Case 6 | Crowdfunding platform | Founder |
| Case 7 | E-commerce product marketplace | Head of Marketplace |
| Case 8 | Crowdfunding platform | Founder |
| Case 9 | Self-service ticketing platform | Business Development Manager (UK) |
| Case 10 | Peer to peer car sharing platform | Vice President |
| Case 11 | Social eating platform | Founder and CEO |
| Case 12 | Couponing platform operating in products, beauty and travels | Country Communications Manager |
| Case 13 | Marketplace for smartphone reparation services | Country Manager |
| Case 14 | Classifieds operating in second-hand machinery industry | Cofounder |
| Case 15 | Job seeking platform | Marketing Manager |
| Case 16 | Platform enabling purpose-built industrial IoT applications to ensure connectivity to devices, applications, and data sources across industrial organizations. | Country manager |
| Case 17 | Dating platform | Country Manager |
| Case 18 | Lead generator platform for service centers | Founder and CEO |
| Case 19 | Caregiving services marketplace | Analytics Manager |
| Case 20 | Meta search engine in travel industry | Senior Growth Strategist |
| Case 21 | Classifieds platform for products | Product director |
| Case 22 | Household services marketplace | Chief architect and Technical Cofounder |
| Case 23 | Metasearch engine for holiday accommodations | Country manager Italy and Portugal |
| Case 24 | Peer to peer car sharing platform | Director of International Expansion |
| Case 25 | People transportation services marketplace | Marketing Manager |
| Case 26 | Cloud-based, open IoT operating platform for plants, systems and machines. | Country Business Developer |

Single semi-structured interviews were carried out with managers within the selected companies and the interview guide was based on the two framework levels developed from the literature analysis. In particular, the questions were aimed at evaluating the configuration of each framework variable and dimension identified thanks to the literature analysis. That ensured that the researchers were able to understand how the firms could be positioned according to the investigated elements. Each interview lasted between one and two hours. To enhance the study's reliability, two researchers participated in the interviews simultaneously and minutes and main messages were sent to the informants for review. The evaluation of the consistency of information has been evaluated through independent coding and cross-checked by the researchers. Moreover, the information gathered has been complemented and

triangulated with secondary sources (such as reports, media interviews and web data) for conducting case studies appropriately, without biases.

The exploratory analysis has been conducted through a pattern-matching logic. Indeed, according to the results of the literature analysis [2,6,40,46], a MSP may perform one or more of three specific functions, namely: (i) matchmaking; (ii) transaction; (iii) maker. The matchmaking function allows the match between two or more users within the platform based on their characteristics and needs. The transaction function provides users with the ability to provide a content (product or service) in exchange for a sum of money through the platform. Finally, the maker function guarantees users the opportunity to create content within the platform. These functions will be detailed in Section 4.1.

Therefore, we have identified seven patterns, each one based on the coverage of one or more of the above-mentioned functions (Figure 2). Each analyzed case was then positioned in one of the 7 patterns. For example, all the cases performing only matchmaking functions have been positioned in pattern 1 while the cases performing both matchmaking and transaction functions have been positioned in pattern 4.

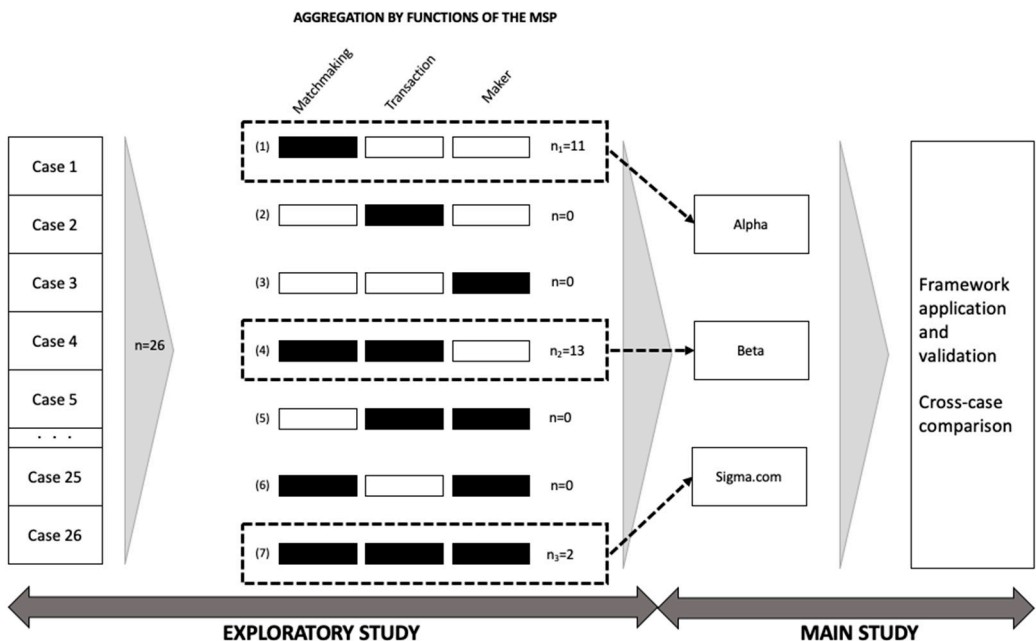

**Figure 2.** The structure of the empirical analysis.

This has been done in order to ensure that the theoretically emerging patterns are creamed off, excluding combinations with no matches in the reality [69]. The exploratory analysis revealed that only three of the seven possible patterns make sense in the DMSP domain (Figure 2).

The main study followed the exploratory study and consisted of three in-depth case studies, one for each of the three patterns identified in the previous phase. The three in-depth case studies are summarized in Table 4.

Semi-structured interviews were conducted with managers covering key executives in marketing and sales functions. Both secondary data and interview transcripts have been used to fill in the developed framework. This analysis allowed to both generalize the framework and to carry out comparisons among the analyzed cases.

**Table 4.** Exploratory case: company descriptions.

| Company | Description | Geographic Coverage | Platform Sides | Role of Respondent |
|---|---|---|---|---|
| Alpha | Italian company providing a solution to help customers and taskers to meet, interact and close good deals. Customers request for price quotation filling out a specific form on the platform with all the detailed information of the service needed. Taskers are able to see this request and, if interested, they pay a fee to obtain the contact information of the customer in order to provide a price quotation. The customer can receive up to 5 price quotations. | Italy | Customers Taskers | 1 interview Role: Company CEO and founder |
| Beta | Multinational French company operating in long-distance ridesharing. Through a large community of users, Beta enables interactions between drivers and passengers willing to go to the same destination and share the cost of the journey. | Europe, Brasil, India, Mexico | Drivers Passengers | 2 interviews Roles: (1) Marketing manager (2) Public relations manager |
| Sigma | Multinational American cloud computing company that provides business software solutions on a subscription basis. The company is well known for its on-demand customer relationship management (CRM) solution offering users with a customer community, developer community and an app exchange marketplace. | Global | Customers App developers | 1 interview Role: Exclusive Sigma app developer |

## 4. A Framework for Characterizing Multisided Platforms

### 4.1. Structure and Methodological Approach

In order to develop the framework for our study, we investigate BM constituents and their potential options [39,70], only considering the relevant elements in the domain of digital multisided platforms.

The framework structure is depicted in Figure 1 and it is hierarchically organized in three levels, namely:

1.  Level 1—Dimensions (D): The first level of the framework consists of six dimensions covering the main features of MSPs. The dimensions have been defined based on the findings of the literature analysis, that have been matched with the BM constituents [39] relevant for MSPs

2.  Level 2—Variables (V): A set of variables has been identified in order to characterize each dimension of the first level of the framework. The variables derive from the literature analysis with some refinement from the exploratory study;

3.  Level 3—Items (I): the third level of the framework encompasses a set of items that operationalize the variables of the second level. They have been defined based on the exploratory study of the 26 MSPs listed in Table 3.

The framework covers all the investigation areas found in the literature (Table 2) except for "Regulation and Antitrust". In fact, this study is not interested in policy, regulation and antitrust or other environmental factors but only focuses on the endogenous characterization of a DMSP.

The third level of the framework translates each variable into one or more measurable items. In particular, there are three types of items, namely:

1.  Binary items—They measure the presence of a specific platform feature, in the platform under investigation, and take the value "yes" or "no";
2.  Configuration items—they can take a value among a set of predefined ones, which represent the possible configuration options;
3.  Open items—they are qualitative items for which it has not been possible to identify pre-defined set of configuration options, so an open description is allowed.

The six dimensions with the respective variables and items are illustrated in the following sections.

### 4.2. Platform Value Proposition

The value proposition indicates the reason why users join the platform [39]. The platform achieves the value proposition enabling specific interactions between the sides and exploiting network externalities. Moreover, the platform may carry out three distinct functions: matchmaking, transaction and maker. The variables that are used to describe this dimension are (Table 5):

1.  Value proposition. The value proposition strongly depends on both the industry in which the platform operates and the services offered [71]. Indeed, an unclear definition of the value proposition may cause the failure of a business since it represents the main pillar of a BM.
2.  Function. DMSPs are diffused in several business sectors and may perform one or more of three different functions, namely: matchmaking, transaction and maker [53,72]. The matchmaking regards the capability to match the demand and offer among the sides. The transaction function refers to the possibility to make a transaction, between demand and offer, with the corresponding payment process through the platform. Finally, the maker function denotes the provision of specific tools or instruments that can be used by the users of a side to create, within the platform, a content to be transacted.

**Table 5.** Variables, items and configuration options for the "value proposition" dimension.

| Variables | Items | Item Type | Options |
|---|---|---|---|
| Value proposition | Value proposition | Open | |
| Function | Function | Configuration | Matchmaking; Transaction; maker |

### 4.3. Platform Sides

The peculiar characteristic of a MSP is the presence of two or more sides [3,46]. This framework dimension aims at scrutinizing the characteristics of each side involved in the platform. The variables that describe this dimension are (Table 6):

1.  Sides. This variable aims at defining how many sides participate in the MSP and their roles [5,10]. Not all the roles pointed out may be covered by the sides because this strongly depends on the functions performed by the platform as well as the industry in which the platform operates;
2.  Segmentation. The platform may create a segmentation of different types of users within each side (e.g., premium users with additional functions or facilitations) [43];

3. Engagement incentives. This variable investigates the presence of mechanisms that incentivize users in the platform to invite others to join [44];
4. Direct externalities. This variable analyses the presence of mechanisms that make more valuable the joining of a potential user in one side based on the number of users already present in the same side [21].

**Table 6.** Variables, items and configuration options for the "platform sides" dimension.

| Variables | Items | Item Type | Options |
|---|---|---|---|
| Sides | Number of sides | Configuration | From 2 to N sides |
| | Sides type | Configuration | Supply; Demand; Peer; Maker; Advertisement |
| Segmentation | Presence | Binary | Yes; No |
| | Segment participation criteria | Configuration | Payment of a fixed fee; Payment of an interaction extra-fee; Achievement of a specific objective; Platform registration |
| | Benefits | Configuration | Enhanced services and/or functions; Enhanced visibility |
| | Benefit standardization | Configuration | Standard; Customized |
| Engagement incentives | Presence | Binary | Yes; No |
| | Reward | Binary | Yes; No |
| | Reward type | Configuration | Amount of money to be spent in the platform (for both users); Amount of money to be spent in the platform (only for the user already present to the platform); Amount of money to be spent in the platform (only for the user invited to join the platform); Reward different from an amount of money to be spent in the platform |
| | Reward setting | Open | |
| Direct externalities | Presence | Binary | Yes; No |
| | Direct externalities characteristics | Open | - |

### 4.4. Platform Revenue Model

The revenue model concerns how economic flows (types, frequency, entity, sides involved) are set in a DMSP. They are (Table 7):

1. Affiliation fees. Affiliation fees are paid by the users to the platform manager, in order to join the platform [3,73];
2. Interaction fees. Interaction fees are paid by the users to the platform manager whenever an interaction is carried out by the platform users [3,60];
3. Financial flows between sides. A financial flow between sides may be present between users of two different sides and it is generally related to a transaction payment for the exchange of a product or a service [74];
4. Referral fees. Referral fees represent economic flows that are given to a specific user of a side as a reward for its specific actions [75].

**Table 7.** Variables, items and configuration options for the "platform revenue model" dimension.

| Variables | Items | Item Type | Options |
|---|---|---|---|
| Affiliation fees | Presence | Binary | Yes; No |
| | Payer | Configuration | [All the sides involved in the platform] |
| | Standardization | Configuration | Standard; Customized |
| | Frequency | Configuration | Una tantum; Regular frequency [specify] |
| | Amount | Open | |
| Interaction fees | Presence | Binary | Yes; No |
| | Payer | Configuration | [All the sides involved in the platform] |
| | Standardization | Configuration | Standard; Customized |
| | Interaction charged | Open | |
| | Calculation | Configuration | Fixed fee per each interaction; Percentage of an economic flow related to the interaction |
| | Amount | Open | |
| Financial flows between sides | Presence | Binary | Yes; No |
| | Transaction object | Open | |
| Referral fees | Presence | Binary | Yes; No |
| | Recipients | Configuration | Sides involved in the platform |
| | Amount | Open | |

*4.5. Platform Control*

Since a MSP enables the interactions between different users, control mechanisms should be set to prevent inappropriate behaviors and actions by the users that can damage the image and reputation of the platform [11,53,55].

The variables identified to describe the platform control are (Table 8):

1. Control mechanisms. The mechanisms arranged by the platform aim at controlling the behavior and the activities of the users as well as the contents provided through the platform [35].
2. Rating and review system. The presence of a rating and review system helps both users in choosing the best match for their need and the platform manager in verifying potential incorrect behaviors [76].
3. Exclusive agreements and contents. The presence of exclusive agreements between the platform manager and users allows the former to provide exclusive services or products so users are forced to join that platform [77].

*4.6. Platform Competition*

This dimension investigates the presence of both inside and outside competition [8]. The variables analyzed are (Table 9):

1. Inside competition. Inside competition is the competition within one side. This generally might occur among the users of the supply side [47];

2.  Outside competition. Outside competition refers to the competitors of the DMSP under study. Competitors could be either MSPs or traditional businesses providing a similar value proposition [77];
3.  Multihoming. This variable evaluates how easy it is to multi-home since the platform manager can make it difficult with some expedients (e.g., investment costs, learning curves), creating an entry barrier towards competitors [47,78].

**Table 8.** Variables, items and configuration options for the "platform control" dimension.

| Variables | Items | Item Type | Options |
|---|---|---|---|
| Control mechanisms | Presence | Binary | Yes; No |
| | Type | Configuration | Identity check; User requirements; Contents (products/services) quality; Respect of the rules of the platform |
| | Timing | Configuration | Ex-ante; ex-post |
| Rating and review system (R&R) | Presence | Binary | Yes; No |
| | Sides involved | Configuration | [All the sides involved in the platform] |
| | R&R direction | Configuration | Unilateral; Bilateral |
| | R&R privacy | Configuration | Public; Partially public; Private |
| Exclusive agreements and contents | Presence | Binary | Yes; No |
| | Side(s) involved | Configuration | [All the sides involved in the platform] |
| | Benefits characteristics | Open | |

**Table 9.** Variables, items and configuration options for the "platform competition" dimension.

| Variables | Items | Item Type | Options |
|---|---|---|---|
| Inside competition | Presence | Binary | Yes; No |
| | Sides involved | Configuration | [All the sides involved in the platform] |
| | Platform manager influence presence | Binary | Yes; No |
| | Platform manager influence type | Configuration | Enhanced visibility respect others users; Showing ratings results; Specific recognitions by the platform manager |
| Outside competition | Main competitors organization model | Configuration | Platform business; Traditional business |
| | Main competitors value proposition | Configuration | Similar value proposition; Partial overlapping value proposition |
| | Main competitors geographical market | Configuration | Same geographical market; Different geographical market |
| Multihoming | Multihoming | Configuration | Allowed; Partially allowed; Forbidden |

### 4.7. Platform Architecture

The platform architecture consists of the infrastructural organization of the digital platform, focusing in particular on technological aspects and the interfaces with the users [2,20]. The variables included in this dimension are (Table 10):

1.  User registration. The registration of the user might be needed or not to join the platform [79–81];
2.  Boundaries between sides. The boundaries between the sides in a platform may be blurred since the same user may belong to one side or to another one based on the role performed in the interaction [43,82];
3.  Versioning and update. This variable aims at understanding in which way the platform updates are arranged by the platform manager and how the versioning is organized [83];
4.  Platform access. This variable aims at investigating which are the "access ways" that the users can adopt in order to interact with other users [84];
5.  Openness. The openness in a platform concerns the freedom, for the users of the sides, to access and modify the source code of the platform in order to enable coinnovation, as well as the freedom to access the data gathered [5,17].

**Table 10.** Variables, items and configuration options for the "platform architecture" dimension.

| Variables | Items | Item Type | Options |
|---|---|---|---|
| User registration | User registration | Configuration | Registration necessary to access; Registration necessary to interact; No registration needed but it allows to benefit from customized services; No registration envisaged in the platform |
| Boundaries between sides | Boundaries between sides | Configuration | Clear distinction between sides; No distinction between sides |
| Versioning and update | Versioning and update | Configuration | Platform versions automatically updated with no charge; Platform versions automatically updated with charge; Platform versions "updatable" with charge; Platform versions "updatable" with no charge |
| Platform access | Web portal implementation | Binary | Yes; No |
| | Dedicated application implementation | Binary | Yes; No |
| | Operating system (app) | Configuration | iOS; Android |
| Openness | Platform openness | Configuration | Closed; Open |

## 5. Main Study: Description of Companies

### 5.1. Alpha

Alpha is an Italian DMSP for professional services, meant to match the demand by private users with the offering by plumbers, photographers, personal trainers, etc. A customer may request up to five quotations for a specific task and (s)he can directly select and contact the preferred professional based on the quotations received. Alpha aims to help companies find new clients without investing in advertisement.

The business was launched in 2013 in four large Italian cities. At the early stages, Alpha was focused only on household services and acted as a commission-based intermediary. However, after six months, the company realized that the success rate of professional taskers was low, and the platform did not get enough revenue. The revenue model thus changed to the current one, in which the professional pays a fee for each quotation made, while the transaction is arranged outside the platform. Currently the company employs about 20 people and has a community of about 20,000 professional taskers.

### 5.2. Beta

Headquartered in France, Beta is Europe's foremost ride sharing company and the largest community in the world for shared trips by car. It enables interactions between drivers and passengers, allowing drivers to "sell" seats in their cars for long distance rides. In this way, the drivers can cover at least their driving costs and passengers can gain access to cheaper travel.

However, the ride price cannot be set freely by the drivers but must be within a minimum and a maximum price, set by the platform, depending on the length of the journey. The booking and payment of the ride is arranged through the platform. The passenger (driver) pays (gets money) every time it rides (drives) and the company takes a 10% cut on average. At the early stages, the company only enabled the matching between drivers and passengers while the payment was carried out outside the platform. In recent years Beta has become an online booking platform where the transaction is arranged directly on the platform. Currently Beta employs about 400 people and operates in 15 different countries, with an average of more than 6 million rides per month.

### 5.3. Sigma

Sigma is a global cloud computing company that provides business software on a subscription basis and it is well-known for its customer relationship management (CRM) solution. The revenue model of the company has not experienced particular changes from the early years and its solution represents a relatively low-risk undertaking, exploiting the so-called software-as-a-service model.

The Sigma website was launched in August of 1999, and a month later, the company had five corporate customers. A radical change occurred in 2006 with the introduction of the multisided platform model. The company decided to open the software, providing several independent developers with the opportunity to build applications to be sold in a virtual store. The store allows the user to customize its experience with Sigma based on its needs.

## 6. Findings and Discussion

This section organizes the findings from the three case studies in light of the framework presented in Section 4 and discusses the configuration of the variables and items.

### 6.1. Value Proposition

The value proposition may be subject to significant changes over time, particularly in the early stages of a DMSP (Figure 3). Alpha has experienced an in-depth change in its value proposition. At the beginning, the platform was configured as a marketplace for professional services, performing both the matchmaking and transaction functions. However, this strategy changed as soon as the company figured out that the services provided by the supply side were very customizable and it was difficult to force a client to pay in advance. Then the company decided to only enable the matching between customers and suppliers and let them arrange the transaction privately, outside the platform. The opposite journey was carried out by Beta, which started with the objective to only match drivers and passengers (matchmaking function). In this way, the platform was able to reach the critical mass of users, achieving a high level of trust. Then, the platform began also performing the transaction function, introducing a payment process that requires passengers to pay in advance for the ride. Differently from Alpha, the services provided by Beta are standard and not significantly customizable by the demand side. The commoditization level of services and products transacted through the platform

fundamentally affects the functions enabled by the platform itself. A DMSP promoting the provision of commoditized services, such as Beta, aims at removing as much friction as possible, enabling the transaction and the related payment inside the platform. In turn, the service requests by the demand side in Alpha might be very varied and customized.

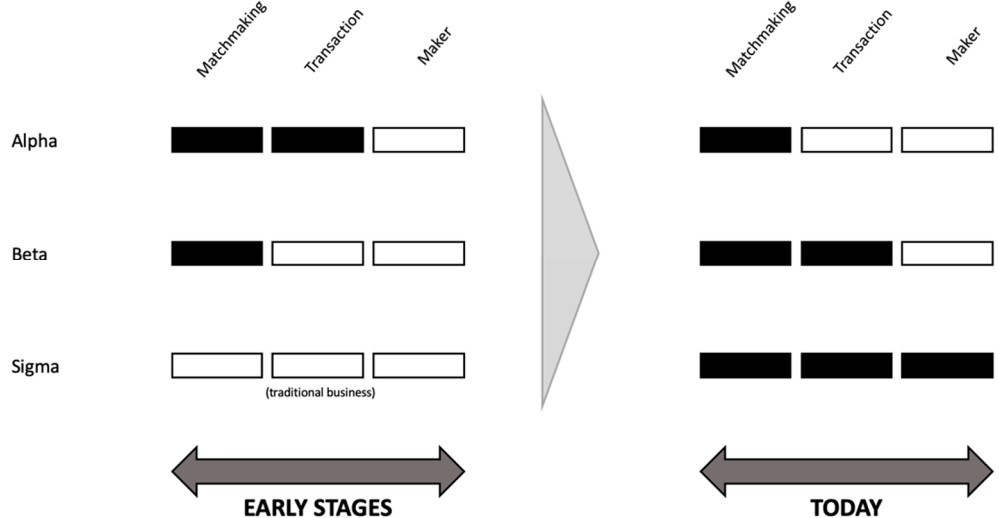

**Figure 3.** Evolution of value proposition and functions in the three case studies.

As regards the "maker function", it strongly depends on the context in which the platform operates and in particular on the contents transacted. This function is generally performed when one-to-many transactions are possible, that is the user can use the same content a theoretically infinite number of times (applications in the virtual store of Sigma). More generally, typical examples in which a maker function is present are those of software, videogames and applications for mobile or desktop operating systems (OS).

*6.2. Platform Sides*

In a DMSP there could be varying numbers and types of sides (Figure 4); there are at least two sides involved, a supply side and a demand side. If the "maker function" is performed, then generally the supply side also performs the role of maker side. In addition, an advertisement side may exist with different effects on the functioning of the platform. On the one hand, it might be a source of revenue for the platform manager; on the other hand, it may create snags and frictions for users in achieving the interactions and transaction with the users of the other sides. Among the three main cases, only Alpha presents an advertisement side promoting product or services related to the businesses in which the professional taskers operate. If the platform performs the transaction process, the platform manager generally manages to achieve enough revenue in order to cover the costs of maintaining the platform. Conversely, if only the matchmaking function is performed, the revenue achievement may be more difficult, so the introduction of an advertisement side might be needed.

Within a side, the platform may create specific sub-groups configured as segmentations. For instance, MSPs may create a "premium segment" to allow users to benefit from enhanced services in exchange for a (higher) fee. In other cases, a segmentation in the supply side might be aimed at enhancing the visibility of a group of users towards the demand side. Another way to segment a side may be related to the achievement of a specific objective of quality concerning the services provided. As an example, Beta drivers who have achieved specific quality targets become "premium users" and can exploit enhanced visibility in the platform.

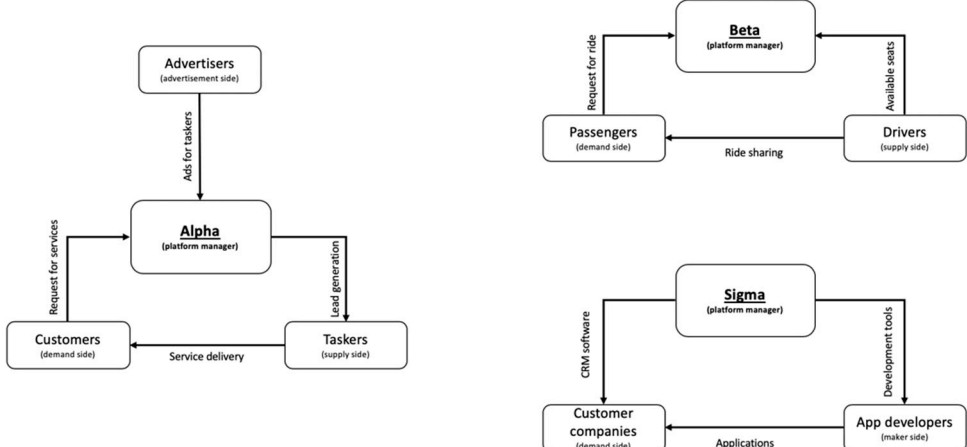

**Figure 4.** Platform sides in the three case studies.

None of the three main cases have engagement incentives currently, even though they had used them in the past. Typically, incentives are set at the early stages of platform inception for increasing the user base. Direct externalities in a DMSP may arise due to the presence of services enabling networking and social dynamics among the users (e.g., web chat) or when the contribution of several users in the same side may increase the probability of achieving a specific result or reducing costs. Beta presents both these aspects because a large number of passengers increases the probability of finding other passengers for a trip and reduces the costs since expenses are shared.

### 6.3. Platform Revenue Model

The type and setting of the revenue streams in a DMSP can be very complex (Figure 5).

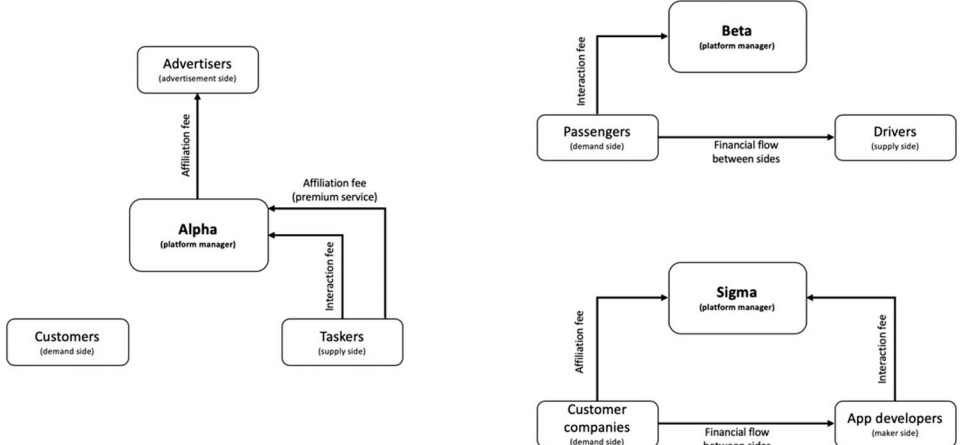

**Figure 5.** Revenue model in the three case studies.

The affiliation fee has to be paid independently from the number of interactions enabled by the platform and it is generally set when it is expected that the user adopts the platform very frequently. In Sigma, the demand side pays an affiliation fee (with regular frequency) in order to get the right to access the platform and download all the suitable applications. This usually happens in platforms performing maker function such as mobile and desktop operating systems as well as in videogame platforms. In other cases, the affiliation fee may be related to a premium service. A well-known example is the "Prime" service by Amazon that lets the users of the demand side benefit from one-day delivery in exchange for an annual fee.

The interaction fee, instead, is required each time an interaction occurs. In Alpha and Sigma the interaction fee is paid by the supply side, while in the case of Beta is paid by the demand side. If the

platform performs the transaction function, the interaction fee is usually a percentage of the amount of money transacted. Conversely, if the MSP has no transaction function, such as in the case of Alpha, the platform may set a fixed fee per interaction achieved through the platform.

Besides affiliation and interaction fees that contribute to the platform manager profit, financial flows between the sides can also be present. That is the case of Beta and Sigma since they perform the transaction function. In Alpha, instead, the transaction is achieved outside the platform and the platform only enables the lead generation.

## 6.4. Platform Control

DMSPs have to implement mechanisms in order to control both the behavior of the users and the contents transacted. A Rating and Review (R&R) system is present in several DMSPs and may be set according to different configurations. In the exploratory study, we found that the rating is general unilateral (from the demand side to the supply one) in B2C and B2B contexts (Alpha and Sigma), while it is generally bilateral in P2P contexts, as in Beta. Furthermore, the ratings achieved by the users of the supply side are generally used by the platform manager in order to check the quality of the products or services provided by the supply side. For example, in Beta, drivers who received a number of bad reviews are automatically banned by the platform manager. The same occurs for professionals in Alpha.

The analysis of the exploratory case studies also revealed that R&R systems are quite common in DMSPs, especially if the transaction function is performed; that is because the platform manager has higher responsibility with respect to the quality of product or services transacted and needs, keeping the trust level high.

Finally, a platform manager may set exclusive agreements with specific users participating in the platform. This choice depends on the industry context. Typically, this happens in DMSPs operating in computer and software industry like videogames, desktop and mobile OS rather than in food delivery where it is possible to set exclusive agreements with specific restaurant chains.

## 6.5. Platform Competition

Depending on the industry in which the MSP operates as well as the strategy adopted by the platform manager, inside competition is often present. This might be due to the price of products and services, such as in Alpha or Beta, or the skills and competences in developing an application as in Sigma. If inside competition is present within the platform, the platform manager may influence with specific mechanisms. For example, it may give the opportunity to show the products or services of a specific user, in the supply side, on the "top of the page" of the results of a search query. This is the case of Beta and Sigma. Another potential way to influence the inside competition is to publicly show the results of the ratings about the users participating in the platform. The ratings are publicly shown in Beta and Sigma. At the same time, the platform manager might influence the inside competition by giving a recognition to virtuous users. Beta drivers who achieve a set target quality level gain a specific acknowledgment visible in the platform.

However, the preliminary study showed that it is possible to have a MSP without inside competition and this happens when the matching between users of demand and supply side is carried out directly by the platform. This is the case of the famous business Uber, which provides a standard taxi service to get passengers from point A to B.

The outside competition concerns the competitors of the platform manager. Competitors might be platforms providing a similar value proposition or traditional businesses that partially cover the offer provided by the MSP in object.

For example, the competitors of Sigma are both MSPs, e.g., CRM solutions with a store where applications are sold and traditional CRM software.

*6.6. Platform Architecture*

Depending on the policy set by the platform, the user might be subject to a registration and this makes the platform manager able to gather information about the user and to carry out analysis about its behavior and preferences. In several platforms the user registration is mandatory and leads to a customized experience based on the user's preferences. For instance, Sigma provides suggestions to the user of the demand side based on the applications previously downloaded. In some cases, the registration can be seen as an obstacle for some users to access the platform.

Another peculiar feature concerning the platform architecture is the boundaries between sides. How much such boundaries are rigid or blurred depends on the type of platform, the industry, its architecture and organization. Therefore, there are two potential configuration options for this variable: (1) there could be different "access channels" for each side involved in the platform as in the case of Alpha and Sigma; (2) there could be only one channel or login access for the users of the platform. In Beta there is a single channel to access the platform. The user can then decide whether to search for a ride or to offer a trip to other users. In P2P DMSPs, the access is generally the same for both the demand and supply side.

Furthermore, the openness level of the platform depends on the operating context of the business. The platform may be open or not because of the presence of specific tools such as APIs and widgets. Most of the MSPs are closed, even though there are exceptions in some platforms performing a maker function such as Sigma where users of the demand side are able to customize the CRM software through coding.

## 7. Conclusions and Limitations

*7.1. Scientific Contributions*

This paper addresses digital MSPs, the ever-growing businesses grounded on digital and internet technologies aimed at enabling specific interactions among different groups of users. MSPs have been investigated in the literature but a lack of a holistic approach to their organizational features is clear. In addition, investigation methods based on empirical analyses (e.g., surveys, case studies) have been rather neglected in the literature. Therefore, this paper attempts to bridge this gap. Researchers provide three main contributions to research on the topic.

First, this paper systematizes the research areas about MSPs, summarizing the main research findings to date. These areas are: network effects, pricing, integration and control, engagement, competition and advertisement.

Second, based on the literature analysis, a hierarchical three-level descriptive framework is developed in order to fill the absence of holistic models to characterize MSP. With the notable exception of the work by Täuscher and Laudien [26] that focuses solely on marketplaces, this is the first model adopting such an encompassing perspective. The literature analysis has provided the elements for the identification of the dimensions (level 1) and variables (level 2) of the framework, while a multiple case study based on the analysis of 26 MSPs has supported its refinement and operationalization (level 3).

The third contribution stands in having actually carried out an extensive empirical research. Indeed, 26 case studies have been performed to define the items of the third level of the framework and their operationalization. Previous research is not fully substantiated by empirical evidence besides the focus on just one single specific aspect of MSPs. In the literature, theoretical studies predominate and the achieved results are generally not empirically tested.

*7.2. Managerial Implications*

This work can support practitioners in organizing and managing a MSP. Indeed, the proposed framework formalizes the key features of digital MSPs to be configured by the platform manager. This is useful to both start-up companies and "traditional" ones willing to move towards a multisided

platform BM. There are in fact several examples of incumbent companies that are seeking to innovate their BM, partially or completely, to remain competitive in the market.

The operationalization of the variables also allows practitioners to visualize the possible options for configuring the different variables. In addition, the framework can be used as a tool by MSP managers, since it helps to describe the current configuration, evaluating evolution over time and potential future development and innovation. Indeed, the literature states that platform businesses can be significantly improved through following a structured innovation strategy [85]. The evaluation of all the dimensions and variables identified in the framework can help in formulating innovation in the whole business model.

Finally, the framework can be used as an assessment tool for benchmarking a company's business with competitors in the same or other industries.

### 7.3. Future Research Trends and Limitations

As with any study, this one comes with some limitations that also pave the way to future research developments. Further research is needed to refine and empirically test the framework. Even though the framework was developed thanks to a multiple-case study analysis, the number of selected cases might be too narrow. Moreover, it is suggested to perform an explanatory survey to test the variables and the theoretical configurations identified in this study as well as the emergence of new variables.

Wider empirical research may lead to determining typical clusters or archetypes of DMSPs, grounded on both theory and empirical analysis. Indeed, commonalities in the configuration of a specific framework variable can potentially highlight specific platform configuration patterns. Moreover, further scientific applications may concern the analysis and identification of relationships among specific environmental features and DMSP configuration. For example, it would be possible to identify potential impacts of outside competition on the organization of the revenue model. Based on the identification of archetypes of DMSP configuration and the analysis of the relationships with external factors, it would be possible to develop prescriptive or normative models that identify the most suitable DMSP configuration for specific contexts.

Finally, the framework can be used as a base to conduct longitudinal case studies concerning a specific MSP. Indeed, the evaluation of the evolution of the configuration of the variables allows one to identify possible innovation patterns in the BM.

**Author Contributions:** All listed authors have made substantial intellectual contributions to the research and the manuscript. Conceptualization, M.A., F.A. and N.S.; methodology, M.A and M.P.; validation, M.A., F.A., N.S. and M.P.; formal analysis, M.A. and F.A.; writing—original draft preparation, F.A., M.A., N.S.; writing—review and editing, M.A and F.A. All authors have read and agreed to the published version of the manuscript.

**Funding:** This research received no external funding.

**Conflicts of Interest:** The authors declare no conflict of interest.

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
