# Peer review of "A Business Model Framework to Characterize Digital Multisided Platforms"

_2199-8531, doi:10.3390/joitmc6010010_

Round 1

Reviewer 1 Report

Dear authors, 

it was a pleasure to read your manuscript “A business model framework to characterize digital multisided platforms”.

The manuscript covers relevant topic on business model innovations in terms of digital multisided platforms.

Some strong points of the manuscript and some areas for improvement are underlined below.

In general, the manuscript is well-organised and contains all the requisite parts. The core message of the paper is clear; the consistency of the paper is obvious.

Abstract. The abstract is well-developed, including research approach, results and contribution.

Introduction. The introduction part covers the importance of the topic. The research gap is clearly presented as well. The presented practical implications of the paper enhance the value of the research. It is really important that the authors underline the practical implications of the work they have achieved. Maybe some deeper overview on previous research could be done.

Background. All subsections cover a large number of previous studies and deeply explain the digital multisided platforms.

Research Objectives and method. Research steps and design are clearly described. However, it would be worth to describe deeply three specific functions (matchmaking, transaction and maker) – page 7. Also it is not enough clear how three in-depth case studies were selected.

Empirical results. The results are clearly described.

Conclusions. The main conclusions are provided as well as some policy implications and future research ideas. The scientific contribution of the paper is also provided.

Please keep in mind that there are some mistakes (for instance line 245).

Reviewer 2 Report

Broad comments.

Thank you for this article. The article is clearly written. It is interesting, as the authors focus from a holistic point of view to multisided platforms (MSP), instead of the reductionistic approach that had been used before to study MSP’s. They introduce a three-level framework for the characterization of MSPs. This three-level framework is based on literature study and a multiple case study. After developing the three-level framework, it is further developed on the basis of three case studies. Although I am quite positive about the article, I think it can still be improved, based on my remarks at ‘specific comments’.

Specific comments.

Line 54: ‘Perspecirves’ must be ‘Perspectives’

Line 100: Does point 4 also belong to the definition?

Line 102: please look at the heading ‘Characterizing aDdigital Multised Plafrorm’

Line 175: ‘3.2. Exploratory Case Study and Framwork Refinement’ must be ‘3.2. Exploratory Case Study and Framework Refinement

Line 188: The authors write ‘Single semi-structured interviews were carried out with managers (...)’. What questions were asked? Maybe the interview questions can be added in an appendix.

Line 195: The authors write ‘Moreover, the information gathered has been complemented and triangulated with secondary sources’. Please add which secondary sources are meant.

Line 197 - 206: I try to understand what has been done during the exploratory research. Is the following line of reasoning correct (otherwise please make it more clear in the text)?: Three specific functions have emerged from the literature analysis, namely: (i) matchmaking; (ii) transaction; (iii) maker. Thus, 7 patterns could be identified (the three mentioned, and the four combinations that can be made on the basis of these three). Confronting these 7 patterns with the cases in table 3 brought forward that there are only three patterns found in reality (‘make sense in the DMSP domain’, figure 2). This question raised because I had some difficulties in line 204 with ‘This has been done (…)’. It seems to me that some lines are missing between 203 and 204.

Line 213: The authors write ‘Single semi-structured interviews were conducted with managers (...)’. What questions were asked? Maybe the interview questions can be added in an appendix.

Line 218: ‘4.1. Structure and Methdological Approach’must be ‘4.1. Structure and Methodological Approach’

Line 245: Does point 4 also belong to the summing up?

Line 264: must 4.3 not been called ‘Platform sides’?

Round 2

Reviewer 2 Report

The authors have elaborated the article further in a positive way on the basis of my questions. They did a good job, and the article is now of a good quality.